# From Trash to Profit: How Packaging Waste Management Has Driven the Circular Economy—An Integrative Literature Review

**Jorge Alfredo Cerqueira-Streit** [1,*] , **Patrícia Guarnieri** [1] , **Luciel Henrique de Oliveira** [2]
**and Jacques Demajorovic** [3]

1   Department of Business, University of Brasília—UnB, Brasília 70.910-900, DF, Brazil; pguarnieri@unb.br
2   Department of Business, Pontifical Catholic University of Minas Gerais,
    Poços de Caldas 37.714-620, MG, Brazil; luciel@uol.com.br
3   Department of Business, FEI University Centre, São Paulo 01.525-000, SP, Brazil; jacquesd@fei.edu.br
*   Correspondence: jorgealfredocs@gmail.com

**Abstract:** *Background:* The COVID-19 pandemic has intensified the need to adopt a restorative and regenerative model proposed by circular economy (CE). *Methods:* This article aims to identify the current status of packaging waste management in the CE context through an integrative literature review using Scopus and Web of Science databases. Using the Bibliometrix package, 162 articles were analyzed. *Results:* A bibliometric overview is presented, including the prominent authors and journals, and most cited articles, techniques and research methods used. Most of the analyzed articles are of the theoretical–empirical, quali–quantitative type, and plastic is the most studied material when a paper focused on one waste item. The contribution of packaging waste management for the transition to CE is discussed, highlighting important actions such as the replacement of materials to increase recyclability, the installation of voluntary delivery points associated with education programs focusing on the environment and support for cooperatives of waste pickers. In addition, a research agenda was developed that highlights the main gaps identified to guide future studies. *Conclusion*: Finally, the managerial contributions of the study are emphasized in particular by providing insight into the implementation of this model of growing international interest.

**Keywords:** circular economy; waste management; packaging; literature review





## 1. Introduction

Municipal solid waste (MSW) is just the "tip of the iceberg" since many materials, water and energy are used to produce what is conventionally called "trash". Some authors [1] are cautious about the classic concept of sustainability that involves the triple bottom line, as they believe that the economic aspect will always receive more attention/investment when compared to the social and environmental pillars. It is believed that from the concern of a product's design, the dynamic balance between the pillars will tend to be more significant. In this way, sustainability does not become mere philanthropy or a research element for the "least bad" economical option, adding value to the three dimensions of sustainability in an integrated manner [2].

The Ellen MacArthur Foundation [3] defines circular economy (CE) as a regenerative and restorative economic system in which resources are kept in use for as long as possible, and waste is minimized. This concept is based on the idea that the traditional model of linear production and consumption (extract–produce–discard) is not sustainable in the long-term, as it generates waste and depletes natural resources. CE proposes a new model in which materials and products are reused, repaired, renewed or recycled, avoiding the extraction of new resources and reducing waste; thus, CE seeks to create economic, social and environmental value in an integrated manner, encouraging innovation and collaboration between companies, governments and society.

CE is an approach that aims to integrate economic development with the preservation of natural resources through new business models and the improvement of production processes to reduce dependence on virgin raw materials; thus, CE proposes to prioritize more durable, recyclable and renewable inputs to extend the useful life of products and avoid their final destination as waste. This approach seeks not only to reduce the environmental impacts of production but also to generate economic and social benefits for companies and society in general, promoting innovation and collaboration between the different participants involved [3–6].

International pacts, such as the European Green Deal or the Sustainable Development Goals (SDGs), have directly influenced how the world perceives and manages waste [4]. By adding the influence of these pacts to the role of the Ellen MacArthur Foundation, the term CE has spread among governments, academic institutions and companies [5–7]. For the transition towards sustainable development to take place, the current economic model, which is based on the linear logic of extract–make–dispose (take–make–dispose), must be replaced by the circular alternative, which includes principles such as those of the 3Rs (reduce, reuse and recycle).

According to the global report on circular economy, there is still a potential for circularity in more than 90% of the materials on planet Earth. By becoming waste, these materials will contribute to problems such as global warming, biodiversity loss, and air, water and soil pollution. Circular economy therefore represents the necessary transformation to accelerate and increase the scale of the application of multipurpose techniques in the use of resources [8].

For this study, "packaging" is the term used to address all materials used for filling, protecting, handling, distributing and presenting information about the goods. The maintenance of physical, chemical or biological characteristics is achieved with the help of packaging. Packaging also contributes to reducing waste throughout the logistics chain (transport, handling and storage) [9].

For various sectors that comprise the recycling chain, the need for the circularity of packaging used in marketing processes gained greater visibility during the COVID-19 pandemic. These are domestic, commercial or industrial materials and contain recyclable materials, such as paper, plastic, glass or metals. The COVID-19 pandemic led to increased recyclable packaging, given the increase in e-commerce purchases. According to a survey by Mastercard SpendingPulse, which has an index that analyzes retail sales, e-commerce sales increased by 75% in 2021 compared to 2020 [10].

Packaging is a major generator of waste and has enormous potential for recovery. A recent paper [11] presented the theoretical motivations for using packaging waste management as a critical element for circular economy, including reducing pressure on natural resources, reducing the carbon footprint, promoting innovation and developing new circular business models. The authors discuss the challenges associated with packaging waste management, such as the need for collaboration between different sectors and the lack of adequate infrastructure. In addition, the authors argue that implementing the circular economy in packaging waste management can have social and economic benefits, such as creating green jobs and reducing dependence on natural resources.

The fact that ecologically correct materials (such as bioplastics) are still not accessible to most of the population has caused this planetary crisis to contribute to the exponential increase in the volume of packaging waste [12]. In addition to the environmental and social advantages, circularity strategies can bring economic benefits [4,6,13].

Because CE emerged from a grouping of previous theories, such as cradle-to-cradle (C2C) [1] and industrial ecology [14], authors such as [2] reiterate the need for more scientific studies in this regard. Several studies have been published, especially in the last decade, as highlighted by [13,15,16], who conducted systematic and integrative reviews of the literature on circular economy. So far, no integrative review study has focused on packaging waste without choosing a specific material; therefore, this research is original and adds to the literature in the area.

The following question was elaborated: What is the state of the art of packaging waste management in the context of circular economy? Based on this question, this study aimed to identify the state of the art of packaging waste management in the context of CE based on an integrative literature review. Following the protocol of [17], systematic and replicable steps were performed to locate, select and filter the relevant literature. The Scopus and Web of Science scientific databases were considered, resulting in 162 analyzed articles. With the support of the Bibliometrix package, it became possible to present the leading bibliometric indicators, in addition to proposing a research agenda and discussing the practical implications of the articles.

The starting point was to consider CE as an alternative model to the current economic system, as there is concern about potential social, environmental and economic impacts on the lifecycle of materials. This article brings direct benefits to academics, as it presents the current state of knowledge and points out directions for future studies. In addition, it also brings contributions to managers interested in the subject by presenting the discussion of the body of knowledge, bringing to light several managerial implications of the CE of packaging.

This study is structured as follows: in addition to this introduction, Section 2 presents the main concepts used, such as the relationship between circular economy, sustainable development and packaging. Section 3 presents the techniques and procedures used to develop the integrative literature review. Section 4 presents and discusses the main results found, as well as the proposed research agenda. Section 5 presents this research's primary results, limitations, contributions and suggestions for future studies.

## 2. Theoretical Background

### 2.1. Circular Economy and Sustainable Development

The 2030 agenda is composed of the Sustainable Development Goals (SDGs) that were prepared by the United Nations (UN) and can be seen as a great action plan for humanity to advance in several areas by 2030. The 5Ps summarize these aspects: people, planet, prosperity, peace and partnerships. Combating extreme poverty, hunger and climate change becomes a priority through strategies that lead to job and income generation through the sustainable management of natural resources [18].

SDG 12, which aims to ensure sustainable consumption and production standards, brings challenges related to solid waste management, including legislation, financing, available information, consumer behavior and governance [19]. To achieve this and other SDGs, in 2015, the European Union launched the CE action plan to "close" Europe's production chains, reducing the volume of waste through the reuse and recycling of materials, among other actions [4].

The European Ecological Pact (Green Deal), signed in 2019, brings the term circular economy as the best method to mobilize the industry to neutralize carbon emissions, reach new markets through the offer of "sustainable products" and create opportunities for the generation of employment [20]. The countries of the European bloc began to adapt their policies to comply with this new agreement [21].

The central non-governmental organization that has contributed to the dissemination of the theme is the Ellen MacArthur Foundation (EMF), and one of its pioneering actions was to take advantage of the World Economic Forum to launch a series of publications "Towards CE" [3,22]. The concept that these publications disseminate refers to a transforming, regenerative and restorative model from the design of [23].

The CE concept has been disseminated as an alternative for maintaining corporations and a competitive advantage. According to [24], the region that stands out for its pioneering spirit and the number of publications is the United Kingdom. The journal that most often publishes articles about CE is the *Journal of Cleaner Production*, which maintains supremacy in other literature reviews involving the subject [15,25–27].

CE and sustainability are different concepts, but they complement each other. Sustainability is a broader approach that seeks to ensure the balance between economic, social

and environmental dimensions, aiming at meeting the present needs without compromising the ability of future generations to meet their own needs. CE, on the other hand, is a more specific approach that focuses on reducing waste and optimizing the use of natural resources by implementing new business models and more efficient production processes [3,11,15,16,28].

While sustainability seeks to balance economic, social and environmental dimensions, CE emphasizes the importance of rethinking how we produce and consume goods and services to reduce waste and optimize the use of natural resources. In this way, CE is a more practical and concrete approach focusing on tangible solutions for the sustainable use of natural resources. At the same time, sustainability is a broader and more abstract approach that seeks a cultural change concerning how society views economic development and the use of natural resources [2,29].

After analyzing 327 articles extracted from the Web of Science and Scopus databases, Ref. [15] warned of the lack of conceptual consensus on the term circular economy and the lack of theoretical basis to treat the topic more scientifically. Even so, it is an economic model with the potential to generate value, strengthen partnerships and collaborative technologies, improve the corporate image and gain in scale.

In a CE, the 3R principles (reduction, reuse and recycling) accompany the product throughout the chain, from extraction to the final consumer and even after consumption, in its environmentally appropriate final destination or return to the manufacturer [30]. To extend the useful life of products beyond these three basic strategies, authors expand to "6R" and even to "10R" [31].

Ref. [32] studied industrial ecoparks as a way of organizing companies capable of contributing to sharing raw materials, energy and waste. According to the authors, during the construction of an industrial ecology model, the "6Rs" must be respected, adding to the first three the logic of recovery, redesign and remanufacturing [32]; therefore, it is evident that the objective of CE is to prolong the useful life of materials, reducing negative externalities (such as pollution and waste) by controlling the flow of materials and energy [33].

In the broader view of the supply chain, there is a combination of loops capable of providing restorative processes that generate benefits for different actors. Integrating processes and fluidity in communication are prerequisites for the circular supply chain's success and consequent waste generation reduction [34].

In a bibliometric study, Ref. [35] points to the proximity of studies that deal with the drivers of CE, the main ones being industrial ecology and the sharing economy. According to [35], although waste management is seen as a reactive measure and is intended to solve the waste problem after its generation, its emphasis is justified in countries where circular economy is not yet consolidated.

One of the main theoretical problems of CE is the difficulty of large-scale implementation. The transition to a circular model requires significant changes in the forms of production and consumption, involving new business models, more efficient production processes and the adoption of new technologies. The implementation of CE involves changing consumer habits and behaviors; consumers need to be willing to opt for durable, repairable and recyclable products, to the detriment of disposable and single-use products [3,15,36].

To spread the theme to industrialists, the Brazilian Confederation of Industry highlights the transition to CE as a business opportunity. Positive consequences can be observed by acting towards the recirculation of materials, such as reducing dependence on virgin raw materials, increasing process efficiency and reducing waste and spending on final disposal [37]. Figure 1 illustrates the methods to minimize disposal and keep the resource in the production cycle.

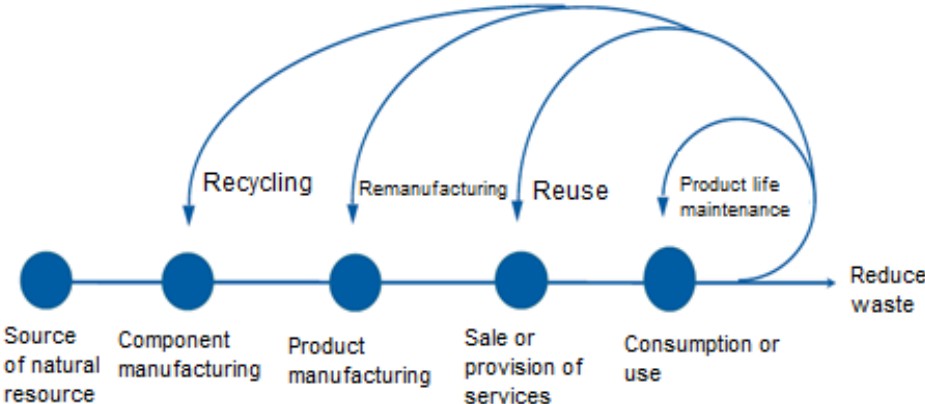

**Figure 1.** Methods of recirculating resources. Source: [37].

Innovations in several fields need to happen to make the transition towards circular economy viable. In many countries, waste management is the crucial point where functional changes must occur to make this transition possible. Technological innovations are important, but insufficient.

Ref. [38] state that technologies are fundamental to connecting actors and facilitating decision-making in favor of a circular supply chain. The Internet of Things (IoT), for example, can expand the ability of companies to operationalize reverse logistics by sharing information in real-time with various stakeholders. In addition to IoT, blockchain contributes to reducing storage costs, which can represent significant gains, depending on the chain. Even so, the authors find that cultural changes are necessary before the implementation of technologies to accept the change to a more circular model.

A change in mindset is also necessary at the producer level (search for materials with excellent recoverability and recyclability), from the distributor (facilitating reverse logistics models) and from the consumer, who needs to start buying remanufactured, refurbished, repaired or recycled items [39].

*2.2. Packaging: Logistical and Marketing Importance and Challenges for Circularity*

Usually made of paper, plastic, glass or aluminum, packaging has the characteristic of being discarded immediately after the product is consumed; thus, the participants in the chain must have environmental responsibility in the project, from production to bottling to distribution. The attention of those responsible involves not only the choice of material, but also the amount used and the adoption of practices that encourage closing the cycle, such as reusing and recycling [40].

Among the most common forms of reuse, packaging reuse and recycling stands out. Reuse presents forms of innovation capable of changing the perception of the product and consequently offering significant benefits to users and companies [41]. Recycling packaging also contributes to the circularity of materials since it transforms the properties of solid waste for full or partial use of the materials.

Packaging management can make a significant contribution to the implementation of CE. Packaging is one of the main generators of solid waste, representing about a third of total urban waste. Proper packaging management can contribute to reducing waste generation through the adoption of durable, recyclable and biodegradable packaging and reducing unnecessary packaging. In addition, packaging management can contribute to creating new business models based on the reuse and recycling of packaging, generating new economic opportunities and reducing dependence on virgin raw materials [11,42].

Although they bring environmental benefits, these strategies are not free of negative impacts. Reuse requires, at a minimum, the consumption of water and the burning of fuel for transportation and the consumption of water and energy for washing. To a greater extent than reuse, recycling also lacks the use of the same natural resources. Even so, it should be emphasized that these strategies cause less damage to the natural environment

than the need for new packaging to be produced from raw material extraction [40]. Figure 2 shows the flow of a package's lifecycle, the potential environmental impacts at each stage and the two reinsertion strategies in the production cycle.

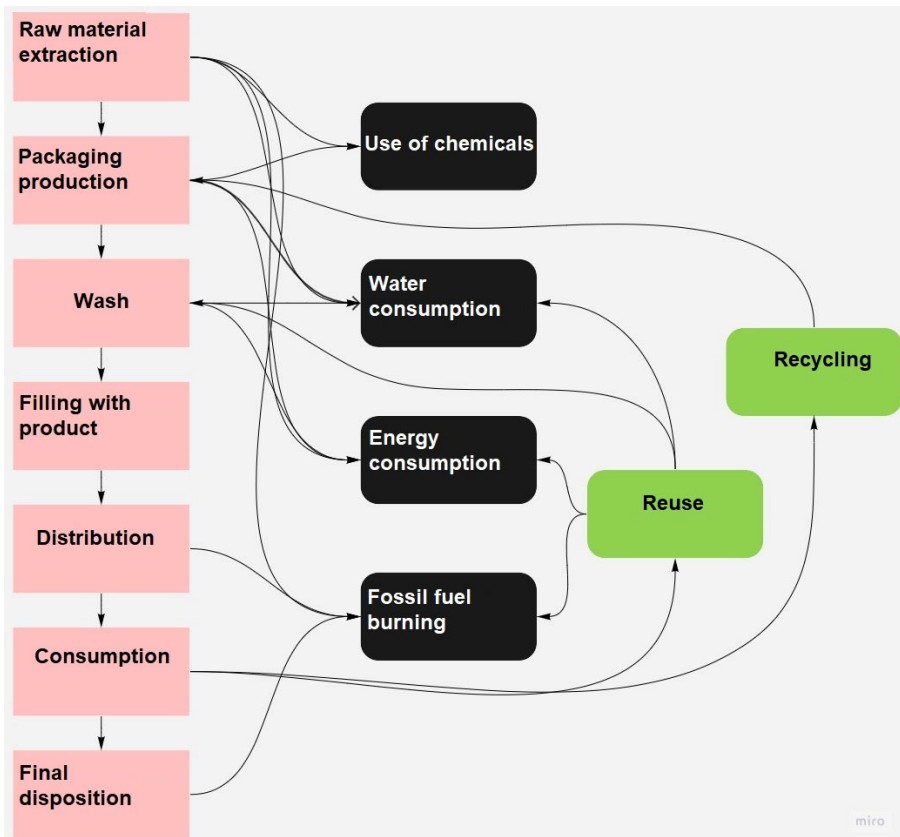

**Figure 2.** Packaging lifecycle and potential environmental impacts. Source: adapted from [40].

In addition to classification according to the main material used, packaging can be divided into primary, secondary or tertiary. Primary packages are those that have direct contact with the product, while secondary serve to group several primary packages. Following the same logic, tertiary packaging protects secondary packaging and its use; therefore, it aims to increase the shelf life of the product and ensure the quality and safety of the packaged/filled products. In this sense, protective layers play a key role in preventing the waste of food and non-food products during handling and transportation [43].

Agreements imposing the fulfilment of shared responsibility for the life cycle of products are necessary, involving all actors in the chain. Seeking a win–win relationship between supply chain actors is critical to building a sustainable supply chain; therefore, the search for intelligent solutions capable of developing cooperation among the actors should be encouraged [44].

### 3. Methods and Techniques of Research

This research can be characterized as applied in terms of its nature, descriptive in terms of its objectives and qualitative in terms of its approach. As for the technical procedure, a literature review was carried out. It should be noted that there are four types of literature review: narrative (traditional), systematic, integrative and tertiary. While a narrative review does not explicitly describe the search, selection and filtering procedures for articles, a systematic review does, focusing only on empirical articles. A tertiary review covers only theoretical articles, and an integrative review is more complete. The integrative review, therefore, explicitly describes the inclusion and exclusion criteria and incorporates both theoretical and empirical articles.

An integrative literature review (ILR) synthesizes the content from the critical analysis of the documents, using systematic and replicable search criteria, selection and filtering of the articles, and may also provide a research agenda full of propositional questions, new taxonomies, classifications and alternative conceptual models that could serve as a basis for new theories [45]. The present work aggregates empirical and theoretical studies; therefore, the authors chose an integrative literature review, thus enhancing the ability to implement practices based on evidence [46].

This study differs from other bibliometric studies and systematic and integrative reviews already published [13,15,16], as it deals specifically with the circular economy of packaging and also considers different inclusion and exclusion criteria.

Ref. [17] present the steps of the RIL performed in this study, which are outlined below according to the protocol:

1. Formulate the research question: What are the characteristics and content of the international literature on packaging waste management and CE?

2. Define inclusion criteria (Filter 1): Using the Boolean AND operator, the selected keywords were "Circular Economy", "Waste management", and "Packag*", incorporating, in this case, "Package" and "Packaging." Only complete scientific articles (theoretical or empirical) published in journals with an impact factor and JCR (Journal Citation Report) in the English language were considered in the scientific databases Scopus and Web of Science. These databases were selected because they cover the leading journals, are evaluated under a double-blind review and have impact factors and a multidisciplinary character. This last factor is significant since several areas of knowledge research the circular economy theme.

3. Define the exclusion criteria (Filter 2): Opinion articles, scientific articles published in event annals (considered in development), errata, books, book chapters, monographs, dissertations, theses, gray literature and white papers, as well as articles published in journals without an impact factor or JCR were excluded. Articles published in languages other than English were also disregarded.

4. Select and access the literature: Scopus and Web of Science (WOS) were the two databases chosen for this research. According to [47], the Elsevier Science (Scopus) and Thomson Reuters (WOS) libraries are similar in size and are continually expanded and updated. Furthermore, these databases are justified because they are concerned with indexing journals with more impact, prestige and influence in their areas [47].

5. Evaluate the quality of reading the proposed topic (Filter 3): Duplicate articles (present in both databases) were eliminated. Then, the titles, abstracts and keywords of the selected articles were read to ensure consistency with the theme.

6. Analyze, synthesize and disseminate the results: The Bibliometrix package was used for this step.

Figure 3 demonstrates the quantitative results found in each of the applied filters.

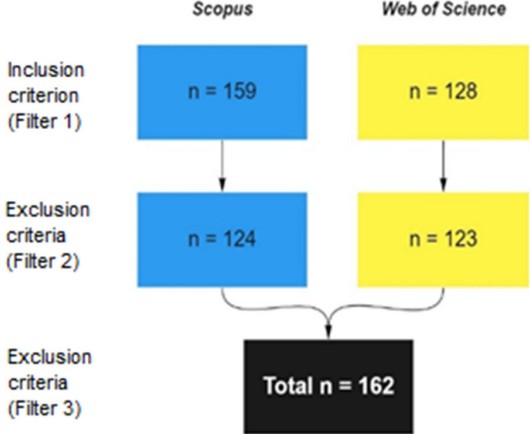

**Figure 3.** Research in the scientific bases and the results found in carrying out the integrative literature review. Source: research results based on [17].

After downloading (.bib files), the articles were imported into the R environment to convert into table format (Microsoft Excel, Professional Plus 2013 version). Then, in joining the two tables (Scopus and Web of Science), 82 duplicate articles were found (present in both databases). Three more repeated articles were found through the column that informed the DOI of the articles (digital object identifier), but with the written title typed differently, which justifies the R software (3.1.4 version) not having considered the different lines. After this "cleaning" of the spreadsheets, 162 articles were selected and formed the sample for this study.

Bibliometrix has open codes programmed in R and RStudio software, which offers tools to perform bibliometric analyses, allows integration with other statistical R packages, and assists researchers in scientific mapping [48]. After filtering, it was highlighted that the analysis of the remaining 162 articles was carried out using tables, graphs and figures to present the leading bibliometric indicators. Furthermore, a content analysis of the articles was carried out to discuss the main findings and propose a research agenda for future studies.

Bibliometrix can integrate the two scientific databases and generate a single database that can be exported to Microsoft Excel (.xlsx); thus, eliminating duplicate articles (present in both databases) was facilitated. It is emphasized that the downloaded files of the articles in the databases must be in ".bib" format, with complete references; thus, the data can be processed via Biblioshiny in the Bibliometrix, and the software automatically generates tables, graphs, similarity matrices and relationship networks, among other forms of bibliographic mapping [48].

## 4. Presentation and Discussion of Results

### 4.1. Overview of Publications

Despite the search covering a vast period, only articles published in 2014 that related to the themes were considered. There was a growing interest among researchers and in journals publishing articles on the subject during that year.

This result is in line with previous studies that state that CE is a topic of increasing notoriety in the scientific community [5,13,15,16,29]. Although 2021 is the year with the most publications (47 of the 162 analyzed), it is worth mentioning that the survey was carried out before the end of 2021; therefore, the number of articles published in 2021 may be even more significant.

Scholars have conducted more research on CE more closely since China and the European Union have included CE in public policy. In China, since 2009, the CE promotion law has been in practice, and deals with the expansion of the recovery of productive resources and the expansion of resource utilization rates, among other measures, according to [35]. According to [33], problems such as air and water pollution, deforestation and soil degradation caused by decades of accelerated economic growth stimulated the Chinese government to pursue the circular economic model.

The results reveal the most significant scientific articles in China and the European Union. Italy tops the list with 30 articles, followed by Austria (21 papers), China (19) and Poland (19). Countries outside these regions also produce research on the subject, albeit on a smaller scale. The Czech Republic (13), United States of America (11), Brazil (8), Russia (5), Australia (3) and Nigeria (3) stand out when investigating packaging waste management in the context of CE. With the aid of the Bibliometrix package of the R software, in which the most potent color is represented where there is more production, Figure 4 illustrates the diffusion of scientific production, which is already present on five continents.

The literature on CE has proliferated; therefore, continuous reviews are necessary to advance the mapping of relationship networks and the analysis of what is being discussed worldwide [15]. However, more important than analyzing the bibliometric characteristics of the articles is the critical advancement in terms of content and related theories, as mentioned by other authors [2,16]. The three-field diagram allows an exploration of complex flows, changing parameters or zooming in and out so that different levels of detail

are exposed. From a minimum set of parameters, the software performs the calculations and simulations [49].

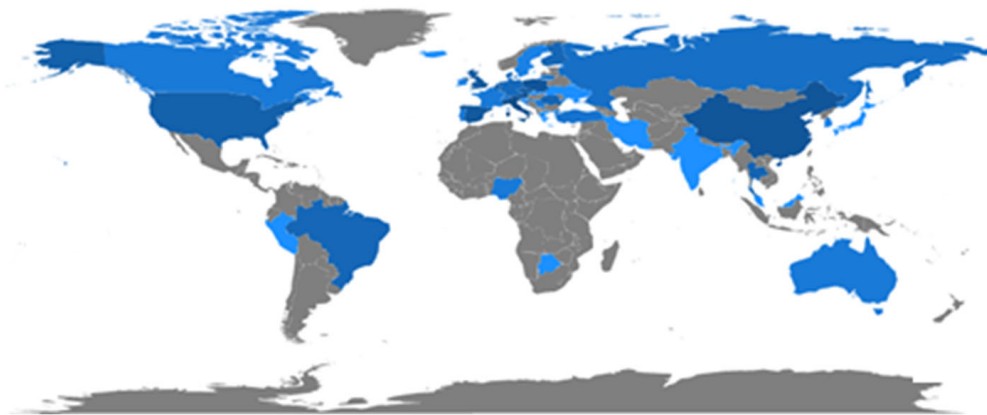

**Figure 4.** Scientific production by country. Source: search results organized by Bibliometrix.

It is valuable to indicate the reviewed articles in a table format to show the reader the journals' quality based on two other metrics. CiteScore is the metric developed by Scopus to measure the impact of scientific journals and the number of citations they obtained in 2018 (data collected on the Scopus website). Table 1 shows the journal data in descending order of the articles analyzed by this research.

**Table 1.** Main journals that make up the integrative literature review.

| Freq. | CiteScore | Citations 2017–2020 | Journal |
|---|---|---|---|
| 29 | 11.5 | 27.451 | *Waste Management* |
| 19 | 14.7 | 23.089 | *Resources, Conservation and Recycling* |
| 13 | 13.1 | 203.300 | *Journal of Cleaner Production* |
| 11 | 4.6 | 2.379 | *Waste Management e Research* |
| 6 | 1.2 | 265 | *Detritus* |
| 6 | 3.9 | 97.894 | *Sustainability (Switzerland)* |
| 5 | 9.8 | 51.386 | *Journal of Environmental Management* |
| 5 | 4.2 | 2.828 | *Journal of Material Cycles and Waste Management* |
| 5 | 10.5 | 210.248 | *Science of Total Environment* |
| 4 | 4.7 | 75.915 | *Energies* |

Source: developed by the authors based on scopus.com/sources.

This review identified Radovan Somplak (from the Institute of Process Engineering, Faculty of Mechanical Engineering, Brno University of Technology, Czech Republic) as the researcher who has contributed the most to publishing articles involving packaging waste management and CE, being present as the author or co-author in 8 of the 162 papers analyzed. Other authors who deserve mentioning include Johann Fellner (Institute for Water Quality and Resource Management, TU Wien, Vienna, Austria), who contributed seven articles to this research in partnership with researchers such as David Laner (5), Emile Van Eygen and Rainer Warrings, who returned results with four items each; and Vladimír Nevrly, who has seven articles involving the researched themes and belongs to the same collaborative network as Radovan Somplak (8) and Veronika Smejkalova (6). Other clusters were present with less participation than the ones mentioned above. Figure 5 shows the surname of the principal author, year and journal containing the articles that received the most citations.

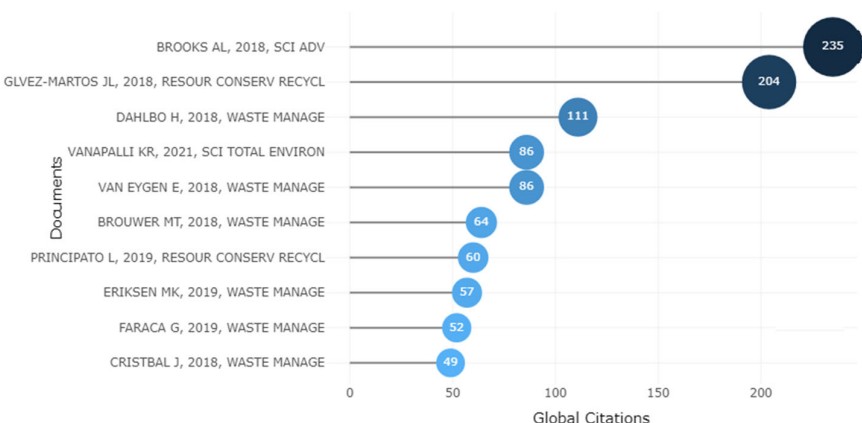

**Figure 5.** Most cited papers. Source: search results, organized by Bibliometrix [14,50–57].

According to the creators of the tool, using Bibliometrix through the R software is still able to show how many times the authors cited a publication present in the review via the DOI (digital object identifier) of the paper and the H index of the authors [48]. Interestingly, the most-cited works were not carried out by the authors who produced the most articles on packaging waste management in CE. Figure 5 shows Ref. [50] as the article cited 235 times, while the authors of [51] were referenced 204 times by other works [52] with 111 articles. Ref. [12] deserves to be highlighted because even though it is a recent article, it obtained 86 citations by the closing of this research.

The types of packaging that are investigated were raised. It was discovered that studies commonly treat packaging in a general way without distinguishing a specific type. Plastics are the most studied materials, followed by food and beverage packaging, which can often mix different materials, such as plastic, paper and aluminum. Table 2 presents the methodological classifications the authors gave, the types of data collection used and the types of packaging that were investigated.

**Table 2.** Methodological profile of publications.

| Search type | Freq. | % | Main Collection Techniques | Freq. |
|---|---|---|---|---|
| Theoretical | 44 | 27.2 | Interview | 16 |
| Theoretical–Empirical | 118 | 72.8 | Survey | 13 |
| Total | 162 | 100 | Elaboration of mathematical model | 4 |
| **Type of approach** | **Freq.** | **%** | End-of-life packaging materials (EOLPM) | 3 |
| Qualitative | 49 | 30.2 | Focus group | 2 |
| Quantitative | 50 | 30.8 | Value-focused thinking (VFT) | 2 |
| Quali–Quantitative | 63 | 38.9 | Workshop | 2 |
| Total | 162 | 100 | Action search | 2 |
| **Type of packing** | **Freq.** | **%** | Main analysis techniques | Freq. |
| General | 87 | 53.7 | Material flow analysis (MFA) | 27 |
| Plastic | 48 | 29.6 | Lifecycle assessment (LCA) | 22 |
| Food | 21 | 13 | Linear regression | 10 |
| Aluminum/metal | 6 | 3.7 | Descriptive statistics | 5 |
| Total | 162 | 100 | Deductive statistics | 3 |
| **Main methods** | **Freq.** | | Analytic hierarchy process (AHP) | 3 |
| Case study | 59 | | Mass and energy balance | 2 |
| Revision | 36 | | Cost/benefit analysis | 2 |
| Gravimetry | 10 | | Six Sigma | 2 |
| Thermogravimetry | 8 | | Technique for order by similarity to ideal solution (TOPSIS) | 2 |

Source: prepared by the authors.

From their methodological classifications, some of the analyzed articles can be exemplified. For example, one of the review articles that made up the sample of this research was prepared by [58]. These authors carried out a documental analysis to investigate the actions implemented by the stakeholders of the plastic chain. Several European directives impose targets for the bloc's countries. Among others, it is mentioned that all plastic packaging sold on the European market must be reusable or recycled in a financially viable way. The work covered several plastic products, such as food containers, beverage bottles and bags [58].

Concerning the quali–quantitative studies, Ref. [59]'s work stands out. Faced with the lack of reliable data in Italy, the authors developed a methodology that allows the creation of an inventory and, with proper statistical automation, can help companies in legal compliance. Then, they applied the method to demonstrate its usability and found that in Italy, there is a lack of tax incentives to motivate companies to reuse materials and, therefore, companies are also not encouraged to measure or report data to control bodies.

To illustrate at least one article that used the lifecycle assessment (LCA) technique, Ref. [60] stands out. The authors compare different strategies for treating plastic waste, such as mechanical recycling, incineration and sending to landfills. Using the LCA allowed the formation of different scenarios for this case (Hong Kong). The results show that mechanical recycling is the best alternative from an environmental and economic point of view; however, they warn of the importance of having recycling industries geographically close to the sources of consumption and sorting so that long distances do not impede the achievement of environmental and financial gains.

This part of the research was concerned with presenting an overview of the main characteristics of the 162 publications considered in this integrative literature review and presents the main methodological aspects of the articles, discussing some selected by alignment with the content of this article. Next, the articles' contents are discussed, focusing on demonstrating how packaging waste management can contribute to CE.

### 4.2. Packaging Waste Management as a Contribution to the Circular Economy

The first article that makes the relationship between CE and packaging waste management, according to the protocols of this research, was [61]. This study investigated three petrochemical industries in Thailand that sought to improve their waste management. In 2010, four years before the study's publication, researchers collected data on the origin, types of waste, quantities and existing management practices.

As much of most petrochemical waste is considered hazardous, these industries also generate paper, plastic, aluminum and metal waste, among other non-hazardous materials that also require proper treatment. Finally, the authors found that the 3R strategy can be used across industries to improve waste management systems. There was a reduction in waste, an increase in reuse and an increase in the recycling rates, which leads to benefits of various natures [61].

Another pioneering work related to the themes of this review was that of [62]. The authors intended to provide information about the behavior of citizens in the municipal management of a small Swedish town (Vellinge). To achieve this goal, they used statistics from a government literature review to obtain the variables that influence disposal, conducted interviews with local managers and applied 117 questionnaires to citizens. According to respondents and interviewees, the decreasing volume of household waste generation is due to a selective collection system that was implemented years ago in Vellinge, in which organic waste was separated from recyclable waste. The authors also found that the socioeconomic variables that most affect waste generation are income, a more expensive waste rate, tourist activity and awareness campaigns. Most survey respondents claimed that the improvement in the municipality's structure was a fundamental factor for them to start separating their household waste, even more important than factors related to environmental concerns.

Investigating post-consumer plastic packaging in a country outside the centers that conduct the most research on CE, Ref. [63] quantitatively analyzed Brazilian plastic waste in the year 2017 with the aid of the material flow analysis technique. In this country, there is a fundamental role for collecting and sorting this type of material: the waste picker. This paper considered the environmental and social benefits of CE when defending the socio-productive inclusion of this category [63].

Although private companies are responsible for most of the formal collection, collectors' cooperatives and the government act in this waste management stage. The authors found that 62% of plastic waste generated is not monitored. The lack of governance and inclusion of informal collectors in the municipal selective collection still makes irregular disposal a problem in the country; therefore, the Brazilian system is similar to a linear system, in which there is no control, revaluation or reinsertion of post-consumer waste. After all, a circular system would not only be able to reduce the volume of packaging waste, but also generate employment and income for historically marginalized workers [63].

The scientific sources that most contribute to the dissemination of the subject are *Waste Management* (Elsevier), with 29 published papers, followed by *Resources, Conservation and Recycling* (Elsevier) with 19 and the *Journal of Cleaner Production* (Elsevier) with 11 articles filtered by this research (Table 1). It is worth describing one of those published in *Waste Management* that makes up the sample of this research and that also used material flow analysis (MFA). In [64], the physical flow of aluminum packaging in Austria was investigated and described, considering the various stages of the process (collection, sorting, transport and destination).

Beverage cans are the materials that most contribute to the total volume of packaging waste (67%), and only 39% of them become secondary aluminum products, having passed through the recycling industry. The most common destination for these packages is sanitary landfills (48%) after being processed in incinerator industries or due to losses due to oxidation. The remainder (13%) is sent to the cement industries since this metal is an essential raw material for cement quality [64].

The authors note that the production of aluminum packaging from secondary raw material (recycled aluminum scrap) should be encouraged. After all, scrap helps in the reduction of energy use, extraction from the natural environment and supply insecurities, positively influencing logistical management [64].

Among the articles published in *Resource, Conservation and Recycling*, the one by Ref. [7] stands out. This article analyzed Brazil's first implementation phase of the packaging sector agreement. Five years after the sanction of the PNRS, the actors involved in the chain signed this pact to comply with the legislation and make the reverse logistics of these packages viable.

The authors note the similarity between the objectives and principles of the Brazilian law on solid waste, enacted in 2010, and what the CE aims at, even though this term is not present in the body of the law. The first phase, which ended in 2017, showed positive results, mainly in terms of the installation of PEVs (voluntary delivery points) and support for cooperatives of collectors of recyclable materials; however, the authors expressed concern about the subsequent phases of implementing the sectoral packaging agreement, given the economic crisis plaguing the country. Due to this crisis, they suggest that future research continue to investigate the opportunities and barriers for Brazil to transition to a packaging CE [7].

It was verified that 8 of the 162 articles selected had the participation of Radovan Somplak, a researcher linked to the Faculty of Mechanical Engineering at the Brno University of Technology in the Czech Republic. The complexity surrounding the topic must be treated quantitatively by this researcher and his colleagues. Ref. [65] used linear programming, a model that involves variables such as the cost of installation, operation and waste treatment. Even non-linear variables are approximated to fit the model, and tests are performed. The objective is that the presented mathematical model supports decision-making for strategies

based on CE; after all, the calculations include reverse flows. It should be noted, however, that they claim that this theoretical model still lacks application in a case study [65].

As shown in Figure 5, Ref. [50] was the most cited article among the 162 articles that comprise the present literature review. Based on secondary data obtained from the Chinese government and UN and World Bank databases, the authors made historical estimates between 1992 and 2017. In December 2017, the Ministry of the Environment declared a ban on the import of plastic waste in China.

China is currently banned from receiving eight types of plastic polymer waste, including PE (polyethene), PS (polystyrene), PVC (polyvinyl chloride), PET (polyethene terephthalate) and PP (polypropylene), among other plastics. With aluminum layers, CDs (compact discs) and DVDs (digital video discs) are still permitted; however, as much as the country has advanced in policies, scholars still project growth in plastic waste by 2035. The pessimistic estimates are due to the lack of indicators to support the market, the low level of state participation and the insistence on the top-down model of executing public policies [50]. This fact has motivated and pressured developed countries that have sent their waste to China to find new solutions.

The integrative literature review on packaging waste management found significant contributions to the management field, with a focus on CE. Among the works analyzed, some examples can be cited, such as the article by [58], who carried out a document analysis on the actions implemented by stakeholders in the plastic chain. The work of [66], who developed a methodology for creating a waste inventory in Italy, and the article by [60], who used the lifecycle technique assessment (LCA) to compare various plastic waste treatment strategies should also be noted.

It is important to emphasize that a literature review should not be limited to reporting what happened, but should seek to make significant contributions to the management field. In this sense, the works analyzed presented relevant gifts, such as demonstrating the importance of mechanical recycling from an environmental and economic point of view and the need to have recycling industries geographically close to consumption and sorting.

It was identified that packaging waste management could contribute to the circular economy, as demonstrated by the works of [56,57], who used the 3R strategy to improve waste management systems, resulting in reduced waste, increased reuse and increased recycling rates; therefore, packaging waste management can contribute to the transition to a more sustainable circular economy.

The contributions presented in this review point to the importance of adopting a broader and more systematic approach to the issue of packaging management and reverse logistics, considering not only the environmental issues, but also the economic and social issues involved. Models and strategies must be developed to increase the recycling rate and reduce inappropriate waste disposal to promote a more sustainable and efficient management of these materials.

From the survey of this review, an agenda of possible research was elaborated with a view to future studies. Table 3 indicates the research gaps identified by other authors.

In addition to showing an overview of publications on packaging waste management in the CE context and discussing the content of some of them, Table 3 presents a research agenda. The purpose of this agenda is to collaborate for the scientific progress of the area to the extent that other researchers can start their investigations based on the gaps the authors pointed out.

The literature review on packaging waste management in the context of CE revealed that CE is a developing field of study, and there are still several gaps that need to be addressed so that the transition to a more sustainable and circular model can be achieved on a large scale. Studies in the area must address these gaps and seek innovative and practical solutions to overcome them.

Table 3. Suggestions for further research contained in the literature.

| N° | Suggestion for Further Research | Reference |
|---|---|---|
| 1 | In developing countries, informal recycling is vital in the return of packaging and the consequent waste recovery. Formalizing activities can lead to sustainable urban development, generating social, environmental and economic benefits. Research is needed to investigate formalization in the recycling chain with the participation of collectors, industry and inspection agents. | [67] |
| 2 | Many municipalities need improvement in their management to properly dispose of food waste and its packaging. Creating a matrix of interests of the actors that participate in the chain helps provide an understanding of the main conflicts. Research should propose different methodologies for implementing policies at the local level. | [68] |
| 3 | Research should contribute to developing innovative packaging to prevent food loss and consumer satisfaction. In addition, tools such as LCA (lifecycle analysis) can help choose the best alternative for final disposal, i.e., incineration, recycling, etc. | [9] |
| 4 | This study presents a framework that unites LCA with Cradle-to-Cradle® Certification and demonstrates the implementation of aluminum cans in a beverage factory. Future research can adapt the framework to other organizations in a case study that demonstrates the implementation of actions that seek eco-efficiency and eco-effectiveness of packaging. | [69] |
| 5 | Poorly developed packaging collection systems are associated with low recycling and material recovery rates. Surveys should verify the input quantities and evaluate scenarios with different systems to estimate the recovery potential and assist municipal waste management. | [70] |
| 6 | From the definition of time and location and with available data, a material flow analysis (MFA) can identify and quantify the generating sources, paths and losses existing in the process. Research needs to evaluate not only the systems using mass indicators, but also the environmental performance of waste management. | [71] |
| 7 | Research should try to compare laws from different regions and countries and their implementation to improve the management of food waste and its packaging. | [72] |
| 8 | Increasing recycling rates does not necessarily mean improving the waste management system. Research should map and help structure a recycling value chain that is capable of incorporating collectors and their cooperatives. | [73] |
| 9 | Containers that are difficult to empty stimulate food waste and make recycling more costly. Research should compare the emptying capacity of different packages to guide ecodesign industry projects and identify the most ecologically friendly option. | [74] |
| 10 | Few papers analyze policy implementation and compare extended producer responsibility in different countries. Research should present the characteristics of packaging waste management systems in different countries, including considering the view of different stakeholders. | [75] |

Source: prepared by the authors.

One of the main gaps is the lack of empirical studies that evaluate the effectiveness of implementing circular economy in different sectors and regions. Although there are several successful cases, few studies have demonstrated the economic and environmental viability of circular economy on a large scale.

Another critical gap is the lack of integration between CE and other fields of study, such as psychology and sociology. Adopting new behaviors and habits by consumers is fundamental for implementing CE; however, few studies have investigated the factors that influence consumption decisions and the adoption of sustainable practices. As CE involves significant changes in the organization of companies and business models, multidisciplinary studies integrating knowledge from management, economics, engineering, health and other disciplines are necessary.

Another gap is the lack of public policies encouraging the transition to circular economy. Although there are some government initiatives in this direction, few studies have evaluated the effectiveness of these policies and propose new strategies to encourage CE.

This study points to its agenda when it indicates the need for theoretical studies that formulate circular economy frameworks based on already consolidated administrative theories. Empirical studies will also be welcomed by the academic community, primarily

if they enrich the debate with a comparison of waste management in different cities (or countries), discussing the importance of public policies for the different levels found. In this way, it is intended to inspire future research.

## 5. Conclusions

Inadequate solid waste management threatens the environment, society and public finances. In response to this concern, the CE concept has been proposed as a viable alternative to reduce the negative impacts of inadequate solid waste management. Although CE has received increasing attention from both academia and public and private decision-makers, there is a continuing need for more scientifically based studies.

Because CE has originated from several other areas, it is not easy to establish a single concept; however, its fundamental principles can contribute to the management of packaging waste. Through an analysis of 162 articles retrieved from the Scopus and Web of Science databases, the main characteristics of these studies were identified, and the relevance of packaging waste management as a contribution to CE was reinforced.

The analysis of the articles reinforced the relevance of packaging waste management as a contribution to CE. During the stages of the process (collection, sorting, transport and disposal), the adoption of practices such as those of the 3R tends to bring socio–environmental, economic and organizational image benefits. In contrast, landfilling waste with potential recyclability is considered the worst treatment strategy. These findings have practical implications for the business and public sectors, including investments in circular technologies to allow for product planning that imagines its reinsertion into the production chain and installing logistical infrastructure to support these initiatives.

This research has practical implications. For developing countries to progress on the subject, it is necessary for institutions participating in sectoral agreements to have more dialogue and effectiveness in their actions. In this way, it is hoped that the private sector, which is mainly responsible for the principle of shared responsibility, will be able to fulfil its role. Still, to bring local implications from the international literature, the installation of PEV (voluntary delivery points) and support to cooperatives of collectors of recyclable materials is recommended.

This paper also contributes to academia, as it sets out guidelines for future research, presenting an overview of publications in the area. This article contributes by introducing the prominent authors, countries and international journals that publish the most research on the subject so that other researchers can broaden the discussion. In addition to the ten points shown in Table 3, there is a lack of circular economy frameworks based on already consolidated administrative theories, as well as applied studies that compare waste management in different regions of the globe from the perspective of circular economy.

As a theoretical contribution, the paper offers insight for managers who seek to build more integrated production chains aligned with the principles of the circular economy. It is necessary to invest in circular technologies to enable the planning of products that can be reinserted in the production chain and the installation of logistics infrastructure. In this way, it is hoped that the private sector, which is mainly responsible for the principle of shared responsibility, will be able to fulfil its role.

Although it contributes directly to other theoretical articles and indirectly to managerial insights, the research has limitations. The use of another research protocol and other scientific databases may lead to different results from those presented, and considering other inclusion and exclusion criteria may lead to other results; therefore, studies that analyze publications in other languages could highlight other factors and consider other scientific bases and standards.

The search criterion exclusively using keywords also restricted the number of articles to be analyzed when compared to the search by publication title (TI) or topic (TS). With regard to the analysis, a limitation of the research is that it only showed some functions of the Biblioshiny for R Studio, while others have the potential for exploitation.

Future studies could consider another integrative or systematic review protocol, such as Methodi Ordinatio, Prockhow C, Prisma and Cochrane. Other works are necessary for the discussion to gain robustness and practical application, and more extensive analyses can shed light on different elements not analyzed in this article. Specific analyses of each type of packaging can also shed light on particularities not addressed in this article. The roles of actors involved in production chains can be investigated through surveys or case studies.

This research has academic implications by exposing guidelines for future research, such as an overview of publications in the area and the prominent authors, countries and international journals that publish the most information on the subject; however, it is vital to highlight the lack of CE frameworks based on already consolidated administrative theories and the need for applied studies that compare waste management in different regions of the globe from the perspective of circular economy.

Despite the limitations, this article can benefit academics interested in this research topic since it identifies the state of the art and gaps that can serve as a starting point for future research. Furthermore, when discussing practical implications, it is intended to contribute to managers who seek to build increasingly integrated chains aligned with circular principles.

**Author Contributions:** Conceptualization, J.A.C.-S. and P.G.; methodology, J.A.C.-S. and P.G.; software, J.A.C.-S.; validation, J.A.C.-S., P.G., L.H.d.O. and J.D.; formal analysis, J.A.C.-S.; investigation, J.A.C.-S.; resources, J.A.C.-S.; data curation, J.A.C.-S., P.G., L.H.d.O. and J.D.; writing—original draft preparation, J.A.C.-S.; writing—review and editing, J.A.C.-S., P.G, L.H.d.O. and J.D.; visualization, J.A.C.-S.; supervision, P.G.; project administration, J.A.C.-S. and P.G.; funding acquisition, J.A.C.-S., P.G., L.H.d.O. and J.D. All authors have read and agreed to the published version of the manuscript.

**Funding:** This research received no external funding.

**Conflicts of Interest:** The authors declare no conflict of interest.

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
