# Peer review of "From Trash to Profit: How Packaging Waste Management Has Driven the Circular Economy—An Integrative Literature Review"

_logistics, 2023_

Round 1

Reviewer 1 Report

The paper represents a thorough analysis of the manner in which concepts related to the circular economy and to packaging waste management and the connection between such concepts has been reflected in the relatively recent speciality literature.

The authors have employed a scientific methodology for carrying out the analysis and the results are interesting.

One point that maybe needs to be addressed is the relative focus on Brazil for various examples and as case studies. So maybe add ”Case study: Brazil” to the paper’s title?

The English language in the paper is very good, there are only occasional problems (like the lack of a plural “s”) that do not in any way diminish the quality of the research.

The English language in the paper is very good, there are only occasional problems (like the lack of a plural “s”) that do not in any way diminish the quality of the research.

Author Response

Dear Reviewer A, thank you very much for your comments. We follow your recommendations to improve our paper. Below are the main changes converging with your requests.

We removed some paragraphs from the introduction and conclusion sections to make it more concise. We mainly removed those paragraphs that referred to Brazil to make the paper more international. We left only a few articles that were applied in Brazil in the theoretical framework, without showing where this work was carried out.

Reviewer 2 Report

Good morning;

Your work is professionally written and structured, and it presents much information that will be a guideline for scholars as well as leaders in the field of circular energy. But to make the tool more enjoyable, it is necessary to review some figures -such as figure 2, 5 and 6) which are not clear and the formatting of paragraph 4.1.

Good luck

Hi ;

The English of this work is good and not difficult to understand.

Thanks

Author Response

Dear Reviewer B, thank you very much for your comments. We follow your recommendations to improve our paper. Below are the main changes converging with your requests.

We have enlarged the font in Figure 2 to make it easier to read.

We created phrases to facilitate the understanding of Figures 5 and 6.

Reviewer 3 Report

The study concentrates on the review of packaging waste management. In order to, conduct the review the authors applied the bibliometric analysis. At the same time, I suggest to make the following improvements:

1. Introduction. The introduction section of the article is too long. I suggest targeting the research problem in this section. Specifically, it could be written why the research was conducted and shortly discussed what was done in the flowing sections. Some of the paragraphs of the Introduction section are irrelevant to the research, e.g. paragraphs about Brazil (lines 87-98). Some parts of the Introduction might be transferred to the Theoretical Background section.

2. Theoretical Background. The Theoretical background section should target similar literature reviews of the problem, their particularities, and methods applied to conduct the research.  

3. Method. Why search of Web of Science was conducted just for keywords? The search for publication title (TI) or topic (TS) in Web of Science might extend the number of publications for the research.

4. Presentation and discussion of results. The results and discussion are very general and don’t use all the potential, which presents Biblioshiny. Despite applying Biblioshiny for R Studio the authors did not even try to build the intellectual structure of the field by applying Co-citation analysis or Bibliographic coupling.

The general conclusion is that the article needs to be improved significantly to be published.

Author Response

Dear Reviewer C, thank you very much for your comments. We follow your recommendations to improve our paper. Below are the main changes converging with your requests.

We removed some paragraphs from the introduction to make it more concise. We mainly removed those paragraphs that referred to Brazil to make the paper more international.

We reiterate the original nature of the paper and the theoretical contribution to the area, as none of the literature reviews published so far focuses on packaging.

We emphasize that the introduction's last paragraph briefly presents the subsequent sections' content.

The search exclusively for keywords was included as one of the limitations of the article since other search criteria could expand the scope.

We also point out, as a limitation, that Biblioshiny for R Studio provides other analyses not presented in this paper.

Reviewer 4 Report

The present article discuss on the role of packaging waste management for circular economy. It has interesting points but in some sections the works is weak and unclear. I strongly suggest the authors to revise the manuscript according to the comments below since I think this work can contribute positively to the scientific community.

It is not clear if this work focus on the Brazilian context or not: the title doesn’t specify it but throughout the text Brazil is usually referred.

Abstract: please do not limit it in explaining the methodology and the content of this review. Add also some results that you want to give.

Please change USW with MSW (municipal solid waste), which is more used.

Line 61: why “reduce, reuse and recycle” is repeated two times?

Lines 87-91: I disagree with this statement but if the Brazilian context says so, please provide specific references.

Line 128: please change the reference style (es. the protocol from Ref. [22]).

I’ve founded several works concerning the circular economy of packaging waste, listed as follow (https://doi.org/10.1016/j.jclepro.2020.120495; https://doi.org/10.1007/s43615-021-00031-2; https://doi.org/10.1016/j.wasman.2019.06.035; https://doi.org/10.1016/j.jclepro.2019.07.057; https://doi.org/10.1016/j.spc.2022.06.005; https://doi.org/10.3390/su14084425). Which are the differences and the novelties of your work? This is missing in the introduction.

In addition, the introduction section is unnecessarily long. Please reduce it and go to the point of the work.

Section 2 is a repetition of introduction. Please condensate the necessary concept in either introduction or theoretical background sections.

Figure 2: can you provide a sharper image please?

Which period of time have you considered? I think a reasonable one is 5 years since it has no sense to consider works older than 5 years ago (the context analyzed will be different from the current one). If you chose another one, please explain the reason.

I don’t understand Figure 5. You stated that your research considered the works that had all the chosen keywords (CE, waste management and packag*). However, from Figure 5 I understand that some works from Italy for example contained only CE as keyword. Can you explain it better please?

Author Response

Dear Reviewer D, thank you very much for your comments. We follow your recommendations to improve our work. Below are the main changes converging with your requests.

We removed some paragraphs from the introduction to make it more concise. We mainly removed those paragraphs that referred to Brazil to make the paper more international.

We split the abstract in MDPI format and added phrases to highlight more results.

Ok, we changed the term to municipal solid waste per your suggestion.

Thank you for letting us know. We deleted the part that was repeated.

Thank you for sending the articles. Soon, we will read them all in-depth and include them in the subsequent discussions on the subject. We take the opportunity to emphasize that many are case studies. At this point, we highlighted the innovative nature of the method used (integrative literature review). We thank you again for your comment. We removed some paragraphs from the introduction to make it more concise.

We have enlarged the font in Figure 2 to make it easier to read.

We write more sentences to facilitate the understanding of Figure 5.

Round 2

Reviewer 2 Report

After reading your article, it is well structured and shows good results. But a few remarks should be made:

1-Change the resolution of figs 2, 5 and 6. They are not Clear

2- Do the same formatting of paragraph 4.1 and before paragraph 4.2

3- It is better if both tables 2 and 3 are on the same page.

Good Luck

Author Response

1-Change the resolution of figs 2, 5 and 6. They are not Clear

Some Figures were created directly by the Bibliometrix, but we could not edit them. Anyway, we send all the Figures in .png to the journal Logistics for possible improvement.

2- Do the same formatting of paragraph 4.1 and before paragraph 4.2

Done. Thank you for the alert.

3- It is better if both tables 2 and 3 are on the same page.

Done. Thank you for the alert.

Reviewer 3 Report

The authors considered my comments and made the corrections.

Author Response

Thank you for the previous suggestions. The paper has been improved.

Reviewer 4 Report

The authors have changed the manuscript according to the comments listed in the previous review round.

Author Response

(The authors gave the same response as above.)

Round 3

Reviewer 2 Report

Hi, 

I think that all remarks have been made but:

- Figure 5 need to be change, to be too clear.

- Put All years of publications in bold.

Good Luck

Author Response

Hi,

Thank you for the notes.

We could not improve Figure 5, so we decided to delete it.

All years of publication in the reference list have been placed in bold.

Best wishes